# Hot-Melt Extrusion Process Fluctuations and Their Impact on Critical Quality Attributes of Filaments and 3D-Printed Dosage Forms

**DOI:** 10.3390/pharmaceutics12060511

**Published:** 2020-06-03

**Authors:** Hanna Ponsar, Raphael Wiedey, Julian Quodbach

**Affiliations:** 1Institute of Pharmaceutics and Biopharmaceutics, Heinrich Heine University, Universitaetsstr. 1, 40225 Duesseldorf, Germany; hanna.ponsar@hhu.de (H.P.); raphael.wiedey@hhu.de (R.W.); 2Drug Delivery Innovation Center (DDIC), INVITE GmbH, Chempark Building W 32, 51368 Leverkusen, Germany

**Keywords:** 3d printing, hot-melt extrusion, process analysis, fused deposition modeling, process optimization, filament fabrication

## Abstract

Fused deposition modeling (FDM^TM^) is a 3D-printing technology of rising interest for the manufacturing of customizable solid dosage forms. The coupling of hot-melt extrusion with FDM^TM^ is favored to allow the production of pharma-grade filaments for the printing of medicines. Filament diameter consistency is a quality of great importance to ensure printability and content uniformity of 3D-printed drug delivery systems. A systematical process analysis referring to filament diameter variations has not been described in the literature. The presented study aimed at a process setup optimization and rational process analysis for filament fabrication related to influencing parameters on diameter inhomogeneity. In addition, the impact of diameter variation on the critical quality attributes of filaments (mechanical properties) and uniformity of mass of printed drug-free dosage forms was investigated. Process optimization by implementing a winder with a special haul-off unit was necessary to obtain reliable filament diameters. Subsequently, the optimized setup was used for conduction of rational extrusion analysis. The results revealed that an increased screw speed led to diameter fluctuations with a decisive influence on the mechanical resilience of filaments and mass uniformity of printed dosage forms. The specific feed load was identified as a key parameter for filament diameter consistency.

## 1. Introduction

3D printing, as a manufacturing technique for individualized drug delivery systems (DDS), has gained a lot of interest in the past six years. Especially, FDM^TM^ offers a high potential for patient-tailored dosage forms regarding dose, shape and release behavior on a low-cost basis [1,2]. Consequently, this is the most explored 3D-printing method for pharmaceutical applications. A drug-loaded filament serves as feedstock material for the FDM^TM^-printer. Since the production of high-quality drug-loaded filaments via hot-melt extrusion (HME) is challenging, the evaluation of filament fabrication and formulation development is of particular interest [3,4,5,6,7,8,9]. In the literature, this has been done mainly in small-scale processes as proof of concept [3,5,6,7,8] and rarely at larger scales [4,9]. At larger scales, other tools related to the filament collection process, diameter adaption and control are mandatory to enable a continuous process.

The filament diameter is a critical quality attribute (CQA) to consider, which is strongly related to the process parameters during HME and the filament collection process [4]. Not only the diameter itself but also the diameter consistency is an important CQA, influencing the quality of filaments and printed dosage forms [10]. Particularly noteworthy factors that are influenced are the printability of filaments, uniformity of mass, content uniformity as well as resolution of printed dosage forms.

The printability is expressed, among other things, in the high mechanical resilience of filaments against longitudinally and transversally applied stress from gears inside the print head, combined with printing accuracy [4,5,8]. Filament sections with smaller diameter tend to be less resistant to mechanical stress. Increased filament diameter potentially leads to clogging inside the printer.

Related to the mass and content uniformity, the diameter homogeneity is especially decisive, since FDM^TM^ printers do not compensate variations. High diameter fluctuations and, thus, dose inaccuracies might occur, which could be distinctive.

These aspects show the importance of a consistent filament diameter, which is often not considered during printing und drug release studies. In previous publications, deviations between 3% and 19% were accepted [9,10,11,12].

How diameter fluctuations relate to the mentioned CQAs, what variations can be compensated, and reasons for their appearance have not been investigated. The hypothesis for the present study was that certain process parameters or combinations of parameters cause pulsatile material transport, resulting in diameter inhomogeneity.

In literature, the target diameter was achieved by stretching and cooling the filament on conveyer belts [13,14,15,16,17]. Monitoring of the filament diameter was done with either calipers [6,11,18], offline after extrusion, at line [9,19,20] or inline [4,14], using diameter measurement modules as depicted in Figure 1. A prerequisite for diameter adaption via a conveyer belt is proper adhesion of the filament to the belt to obtain a reproducible and representative diameter. Additionally, a lopsided position of filaments in the laser unit of the measurement device due to movement on the belt distorts the diameter data. Therefore, a winder should be implemented to potentially optimize the filament fabrication. A feasibility study was conducted to ensure reproducible haul-off of a drug-free formulation and representative inline diameter measurements. Subsequently, a systematic HME process analysis regarding the influences of the screw speed, powder feed rate (PFR) and barrel filling degree on material fluctuations and filament diameter variations was performed. Finally, the impact of potential diameter inconsistencies on the mechanical resilience of the filaments and uniformity of mass of the printed objects should be investigated, identifying acceptable diameter fluctuations and optimizing the HME process for filament production. The mentioned aspects are important to evaluate, since better knowledge will make it easier to follow a quality-by-design approach for filament manufacturing and 3D-printing of DDS.

## 2. Materials and Methods

### 2.1. Materials

A powder mixture of 82.93% (w/w) ethyl cellulose (EC, Aqualon^®^ N10, Ashland, KY, United States) and 16.67% (w/w) hypromellose (HPMC; Metolose 60SH 50, Shin Etsu Chemical, Tokyo, Japan) was used for the HME of filaments. To ensure consistent powder feed, 0.4% (w/w) fumed silica (Aerosil^®^ 200 VV Pharma, Evonik, Darmstadt, Germany) was included. As liquid plasticizer, triethyl citrate (TEC, Citrofol AI Extra, Jungbunzlauer, Basel, Switzerland) was added during extrusion at a concentration of 10% (w/w). Commercial PLA filament was purchased from Prusa Research (Prague, Czech Republic) for comparison purposes.

### 2.2. Methods

#### 2.2.1. Production of Filaments via HME

Batches of 600 g of powder constituents were blended in a Turbula^®^ mixer (T2F, Willy A. Bachofen Maschinenfabrik, Muttenz, Switzerland) for 30 min. The powder blends were fed gravimetrically (KT 20, Coperion K-Tron, Niederlenz, Switzerland) at 5–10 g/min directly into the extruder barrel. Extrusion was performed with a 40D co-rotating twin-screw extruder (Pharmalab HME 16, ThermoFisher Scientific, Waltham, MA, United States) through a 1.85-mm die at 190 °C barrel temperature. The screw configuration, equipped with two kneading blocks (KB, 4 × ¼ D 30°, 4 × ¼ D 60°), and the barrel setup can be found in the Appendix A.

The liquid plasticizer was fed via a pre-calibrated micro annular gear pump (MZR 7205, HNP-Mikrosysteme, Schwerin, Germany) to enable a constant liquid flow. Depending on the applied PFR, the liquid feed rate (LFR) was adapted to obtain the filament composition of 74.62% EC, 15% HPMC, 10% TEC and 0.38% (all w/w) fumed silica. Process parameters were varied systematically for the winder implementation and process analysis (Table 1). Behind the extruder outlet, filaments were cooled. Filaments were stretched via a counter-rotating belt haul-off unit of a winder (Model 846700, Brabender, Duisburg, Germany) to obtain the target diameter of approx. 1.75 mm. The haul-off speed was adapted to the applied PFR to keep the mean filament diameter constant.

For the feasibility study, filaments were collected on a 125-mm spool with a traction of 4.5% (percentage of maximum engine power) of the winder unit. The process setup is displayed in Figure 1B. Sampling of filaments was done over ~300 s for each setting after reaching process equilibrium, indicated by constant pressure at the die and power consumption. Process data were monitored with a frequency of 1 Hz using an in-house-written LabVIEW 2015 application (SP1, National Instruments, Austin, TX, United States) application.

#### 2.2.2. Inline Diameter and Ovality Determination of Filaments

The filament diameter was monitored using a laser-based diameter measurement module (Laser 2025 T, Sikora, Bremen, Germany). The diameter of the cross-sections was recorded in three different directions with a sampling rate of 1 Hz. The smallest measured diameter was used for data evaluation and calculation of the cross-sectional area for the determination of the Young’s modulus (YM). The ovality of filaments was calculated as difference between the maximum and minimum determined diameter per second. The measurement module was placed between the cooling section and the haul-off unit of the winder (Figure 1B).

#### 2.2.3. Offline Diameter Determination of Filaments

For the evaluation of potential deformation and elongation of filaments due to the applied tension on the winder unit, diameter and ovality were measured offline again after initial winding. Filaments were uncoiled using the same haul-off speeds as during extrusion.

#### 2.2.4. D-Printing of Commercial and In-House Produced Filaments and Determination of Uniformity of Mass

3D printing of objects was performed using an FDM^TM^ printer Prusa i3/MK3 (Prusa Research, Prague, Czech Republic) equipped with a 0.4-mm nozzle. Test objects were printed using commercially available and in-house-produced filaments with a layer height of 0.1 mm, a rectilinear infill (90°) and a printing speed of 15 mm/s. The nozzle temperature for EC/HPMC-based filaments was 180 °C, and for commercial PLA filaments it was 210 °C. The build plate was heated to 90 and 60 °C, respectively, to ensure adequate layer adhesion. The geometry was designed using the computer-aided design software Autodesk Inventor Professional 2018 (Autodesk, Mill Valley, CA, United States). The obtained *.stl*-file was converted into a G-code via Slic3r Prusa Edition software (Prusa Research, Prague, Czech Republic). The mean filament diameter varied between 1.75 mm (commercial PLA filament) and 1.78–1.82 mm (filaments based on EC/HPMC), as different process parameters were applied. The diameter in the Slic3r software was adjusted for every filament batch during printing. Twenty rectangular shapes (3 × 15 × 7.5 mm) per batch, as depicted in Figure 2, were printed from filaments with an infill of 85% and tested according to Ph. Eur. 2.9.5 “Uniformity of mass for single-dose preparations”. To ensure representative sampling, printing within one filament batch was conducted, using arbitrary sections from the beginning, middle and end of the extrusion process. Printed test dosage forms were weighed precisely using an analytical balance (Sartorius CP1224, Goettingen, Germany).

#### 2.2.5. Mechanical Properties

The mechanical properties of filaments produced at different screw speeds were determined using a texture analyser (TA.XT plus, Stable Micro Systems, Godalming, United Kingdom) one day after production. The testing regime was performed according to Korte and Quodbach [4]. A tensile test (*n* = 6) and a three-point bend test (3PBT; *n* = 6) were applied for the determination of the YM and distance at break, respectively. Settings as well as data evaluation were adopted from [4,5]. The 3PBT was performed with a 25-mm gap and a test speed of 10 mm/s. For the tensile test, extrudate sections of 20 mm length were clamped between two grips with a torque of 1 Nm. The elongation and corresponding forces were measured using a test speed of 0.01 mm/s. The YM was calculated according to DIN EN ISO 527-1 as the slope between 0.05 and 0.25% elongation of obtained stress–strain curves.

#### 2.2.6. Determination of Specific Feed Load

The specific feed load (SFL) is a measure of throughput characteristic, which is defined as a dimensionless number according to Equation (1) [21],
(1)$$SFL=\frac{\dot{m}}{\rho \;\cdot \;n\;\cdot \;D^{3}}$$
where $\dot{m}$ is the total gravimetric throughput (PFR + LFR), ρ the true density of the formulation, n the screw speed and *D* the internal barrel diameter. The true density of filaments was determined using a helium pyknometer (AccuPyc 1330, Micromeritics, Norcross, GA, United States) in triplicate.

#### 2.2.7. Statistical Analysis

The interquartile ranges (IQRs) of diameter and pressure data were chosen as markers in this study, as they can be calculated without underlying assumptions of the distribution function and are robust to outliers.

The distribution of the test statistic of interest (the difference in IQR) is not known and no conventional statistical test could be used. Hence, the statistical significance of differences in IQR was tested using a bootstrap approach. The datasets of interest were merged and bootstrap samples taken with replacement from the combined set. This bootstrap sample was divided into two parts and the IQR for each part was calculated. Subsequently, the difference in IQR was calculated. By repeating this B = 100,000 times, a distribution of all possible differences in IQR—under the assumption that the two datasets were from the same population—was generated. Comparing the observed difference in IQR with these distributions allowed for the estimation of a *p*-value, i.e., the probability to observe the given difference, even though the datasets were from the same population. The bootstrapping procedures were performed using R (Version 3.6, R Core Team). Means of datasets were compared using conventional t-tests (OriginPro 2019, OriginLab Corporation, Northampton, MA, United States). 

## 3. Results and Discussion

### 3.1. HME Process Setup

The first part of the study aimed to optimize the HME process setup to analyze the influence of process parameters regarding diameter variations. In order to circumvent the mentioned issues, the extrusion line was expanded by a winder, which consisted of a haul-off unit, a traverse and a winder unit. In Figure 1B, the changed process setup is depicted. The haul-off unit consists of counter-rotating conveyer belts. The speed can be adapted precisely to obtain the targeted diameter. After the haul-off unit, filaments were wound up via a rotatable spool, which is coupled to the set haul-off speed and traverse. Implementation studies were performed to examine the winding capability of the developed formulation. A potential elongation and/or oval deformation of filaments due to the applied tension caused by the winder unit was tested by measuring both diameter and filament ovality offline after the winding process again. Simultaneously, it was tested whether a reproducible and steady stretching of filaments by the haul-off unit was possible since this was an observed problem for the initial setup (Figure 1A) due to insufficient adhesion of the filament on the conveyor belt.

For the feasibility testing, extrusion settings listed in Table 1 (T1 and T2) were applied. Two different PFRs (5 and 10 g/min) were tested as the haul-off speed must be doubled to achieve the same diameter. When the haul-off speed is doubled, the available time for solidification on the belt is halved, which might lead to filament deformation during the winding process.

As exemplarily depicted in Figure 1C, produced filaments showed a good winding capability without breakage. Consistent and reproducible diameters for both filaments (T1 and T2) were obtained (Table 2).

In Figure 3, the results for the evaluation of filament changes caused by the winding process are shown. The inline- and offline-determined diameter and ovality as functions of extrusion time are plotted. Using a PFR of 5 g/min, no significant difference in diameter after the winding process was observed (inline = 1.781 ± 0.032 mm; offline = 1.783 ± 0.032 mm; *p* = 0.455). The ovality after the winding process (0.028 ± 0.005 mm) is significantly higher compared to the inline determined ovality (0.024 ± 0.003 mm; *p* << 0.01), which is very likely a measurement artifact caused by a lopsided filament position inside the laser unit, as it was not under full tension during offline feeding. Some deviations are present due to movement while uncoiling the filament from the spool. This is especially visible in the first 20 s of the offline measurement (Figure 3, left). Generally, the ovality is small. It can be assumed that the filaments are almost completely round and have not been deformed considerably by the winder.

Applying a PFR of 10 g/min, a significant decrease of diameter from 1.798 ± 0.031 mm to 1.774 ± 0.032 mm after the winding process was observed (*p* << 0.01) as depicted in Figure 3, right (dark grey vs. light grey curve). The difference was likely caused by increased elongation due to the applied tension of the haul-off or winder unit induced by the decreased solidification time. Plastic deformation related to ovality was difficult to interpret, since small movements during the offline measurement occurred. The mean values and standard deviations were the same for the inline and offline measurement (0.025 ± 0.005 mm; *p* = 0.445), although the fluctuation of the offline measurement was higher.

However, no remarkable change in diameter and ovality was found. The setup was considered suitable to generate reliable data for process analysis. Diameter variations could now be clearly assigned to extrusion-related process parameters. If higher throughputs are used, a more efficient cooling section, e.g., using fans, might be necessary to avoid filament changes. The implementation of a capable winder is especially beneficial with regard to continuous filament production.

### 3.2. Process Analysis

#### 3.2.1. Correlation of Material Pressure and Diameter Variations

Filament diameter homogeneity is an important quality attribute to ensure printability and dosing accuracy. It was reported that variations higher than 1.75 ± 0.05 mm are undesired [22]. Prasad et al. [9] investigated the HME process for HPMC-based formulation to find an operating window where the screw speed, temperature, die diameter and drug load was assessed related to material pressure, torque and filament diameter. The organoleptic appearance was considered additionally. However, the impact of the different settings on material pressure and diameter variations along with CQAs of printed dosage forms was not assessed. Korte and Quodbach [23] systematically evaluated the effect of PFR, screw speed and conveyer belt speed on the filament diameter and also on the mass of printed dosage forms. A significant effect of PFR and the conveyer belt speed on the mean diameter of filaments was found. The variation of diameter and its impact on the mass variation of printed dosage forms was not investigated.

Preliminary extrusion experiments showed that fluctuations of the recorded material pressure are reflected in diameter variations of the filaments, as exemplarily depicted in Figure 4. This has already been demonstrated in the work of Korte [23]. Pressure fluctuations occur because of material accumulations in front of KB and the die. This leads to pulsatile melt conveyance, as a minimum amount/pressure is necessary to force material through the KBs [24]. Consequently, the material amount which passes the die varies and, with it, the diameter. To what extent these fluctuations occur is highly dependent on the viscoelastic behavior of the used material and die swell [25]. In this study, an impact of inhomogeneous powder feeding could be excluded (data not shown). Pressure fluctuations were observed to depend on process parameters. Accordingly, a rational process design was applied to determine the influence of PFR and screw speed on diameter variations. Three different PFRs at a screw speed of 40 rpm and five different screw speeds at a PFR of 5 g/min were tested (Table 1).

#### 3.2.2. Influence of Powder Feed Rate

In Figure 5, the results of the influence of different PFRs (5, 7.5 and 10 g/min) on material pressure (as measure for material conveyance) and filament diameter are displayed as box plots. The IQRs (box width) were used to evaluate the variations. The material pressure increased with increasing PFR as expected, as the barrel filling degree was increased (Figure 5A). The IQRs were comparable. A sufficient amount of material was present to convey the melt uniformly through the KB and the die, resulting in similar diameter variations (Figure 5B).

The differences on diameter variations (IQR), depending on the set PFR, were negligible and relatively low (Table 2). It was revealed that in this operation window the PFR had no considerable effect on material pressure fluctuations and thus diameter inhomogeneity. This result was unexpected and will be explained in Section 3.2.4.

#### 3.2.3. Influence of Screw Speed

Regarding the influence of the screw speed, different results were obtained (Figure 6A,B). An increasing screw speed resulted in lower material pressure because of a decreased barrel filling degree and the potential shear thinning of used polymers. With increasing screw speed, the IQR of the material pressure was increased. This had a distinct influence on diameter variations. Diameter fluctuations were significantly higher (IQR = 0.027–0.082 mm), especially above 40 rpm (*p* < 0.01), although the mean value was similar in all cases (Table 2). A likely reason is that at higher screw speeds the barrel filling degree is lower, resulting in more pulsatile melt conveyance through the KB and die. A faster removal of lower amounts of material takes place and the material held up in front of the KBs until enough material accumulated to build up the necessary pressure to push the melt along the barrel. At lower screw speeds (20–40 rpm) and thus higher barrel filling degrees, the diameter variations were lower (IQR = 0.027–0.044 mm), as enough material was present to convey the formulation more homogenously with less fluctuations. Filaments produced with 20 rpm and 5 g/min showed the lowest diameter variations.

Compared to commercial filament, the self-extruded filament is less homogeneous. The PLA filament has a very low IQR of 0.005 mm (1.756 ± 0.004 mm, CoV_diameter_ = 0.21%; Figure 6B), whereas the variation of self-produced filament showed a 5-fold higher IQR of 0.027 mm (1.782 ± 0.019 mm, CoV_diameter_ = 1.07%).

It has to be considered that commercial filaments are usually produced with single screw extruders with a rather low mixing capacity because it is not required to incorporate materials with a uniform distribution. Consequently, pulsatile material transport and resulting diameter variations are minimal, as depicted in Figure 6C, where the PLA filament diameter is plotted as function of time. In pharmaceutical processing, an extensive, dispersive and distributive mixing is necessary at controlled conditions. This is only provided by twin screw extruders (TSE) [26]. Additionally, process ability is highly dependent on material characteristics, like rheological properties, molecular weight, particle size distribution, etc. Thus, diameter variations can be caused by the material itself but also by the selected extruder type.

A homogenous material throughput along the barrel is expected to be decisive for diameter homogeneity. With decreasing screw speed the barrel filling degree is increased, resulting in the abovementioned lower filament diameter fluctuations. The data from different PFR indicates that the fill level of the barrel was high enough in all cases to obtain homogeneous filaments (Section 3.2.2). In the following, it should be investigated if lower filling degrees with different PFR/screw speed combinations affect diameter homogeneity.

#### 3.2.4. Influence of SFL on Diameter Homogeneity

To investigate the influence of the barrel filing degree, the SFL was calculated according to Kohlgrüber and Wiedmann (Equation (1)) to enable a comparison with other formulations and (potentially) equipment.

In Figure 6B, the SFLs for the respective applied screw speed are listed. The data indicates that if the SFL is low (<0.03), the material transport inside the barrel is more pulsatile, leading to larger diameter variations. If the SFL is higher (>0.03), diameter variations decrease distinctly, likely because of a more homogeneous melt conveyance, independent from the set PFR–screw speed combination. It should be investigated whether the same diameter quality can be achieved by keeping the SFL constant at different settings. In this study, the highest applied SFL was 0.059 and the lowest one 0.02. Both SFLs were investigated at two different PFRs (Table 1). Four different setting combinations were applied in total. Therefore, the PFR was set to a low level (LL) with 5 g/min and to a high level (HL) with 10 g/min. The respective predefined SFLs were reached by applying different screw speeds (20–120 rpm). The IQR was used to evaluate variations.

In Figure 7A,B, the results for the investigation of the SFL are displayed. At a low SFL of 0.02 the pressure data for both applied setting combinations was comparable (mean = 9.7 bar and IQR = 1.4 bar at 10 g/min and mean = 10.0 bar and IQR = 1.9 bar at 5 g/min; Figure 7A). For the high SFL of 0.059, IQRs were also similar (1.3 and 1.2 bar). Even though the barrel filling was the same, the mean material pressure deviated (14.9 vs. 19.6 bar). This was not expected and might be caused possibly by a faulty calibration of the pressure probe at different temperatures. Although the pressure IQRs for the different SFLs did not differ a lot, the influence on diameter fluctuations was distinct (IQR = 0.082 mm at a low SFL and IQR = 0.027 and 0.035 mm at a high SFL). A limitation of the conducted experiments is that the evaluation of the absolute values of the material pressure might be not valid. Figure 4 shows that the fluctuations of pressure data depended on the set screw speed. At 60 rpm, pressure data appeared as a non-fluctuating line as sampling took place at the same screw position (sampling rate 1 Hz). In contrast, at 40 rpm, a signal oscillation was observed, which was not caused by fluctuating material transport but sampling at different screw positions. With the applied screw speed of 40 rpm, only 2/3 revolution per second occurred. Consequently, the initial position was only reached after every third second, resulting in fluctuating data.

Furthermore, the device-related low sampling rate (1 Hz) was probably not sufficient to obtain complete pressure information. The signal to be sampled (material pressure) contained frequency components that were higher than half the sampling frequency, which leads to an artifact, known as Alias effect in signal analysis or computer graphics [27]. This discriminated the output signal, resulting in incorrect amplitudes and apparently lower frequencies. The pressure minima and maxima are then underestimated [28]. The obtained material pressure trends still convey important information about variations, as shown in Figure 4, but absolute values cannot be discussed with high confidence due to the aforementioned factors.

However, the stated hypothesis of the SFL as a key parameter for diameter variations could be confirmed (Figure 7B). For the first time, it could be demonstrated that the same diameter quality was achieved at a given SFL, independently of the PFR and screw speed combination. This is true at least in the investigated operation window.

At higher SFLs and increased barrel filling degrees, filaments with less fluctuations in diameter (IQR = 0.035 and 0.027 mm) compared to lower SFL (IQR = 0.082 mm both) were obtained. The higher the SFL, the more even the material transport is, resulting in a uniform filament diameter independent from the extrusion parameter combination.

For the SFL of 0.02 the mean diameter was slightly higher (1.821 ± 0.062 mm) when 10 g/min were fed compared to the batch where 5 g/min was used as the PFR (1.777 ± 0.064 mm), although the same haul-off speed was applied. This was likely due to the combination of an increased shear stress at the higher PFR with the additional high screw speed of 120 rpm, resulting in higher die swell and thus a larger diameter [21]. The hypothesis was further supported by plotting the diameter IQR as a function of SFL (Figure 7C). A material dependent threshold of 0.03 as SFL was found. Below this value, a sharp increase in diameter variation was observed. This explains why no influence of the PFR was found in Section 3.2.2, as the SFL was ≥0.03 for all combinations. For the SFL of 0.059 the lowest diameter variations were found. To underline this assumption, further analysis, including other formulations and extruder systems, is desirable.

The presented results are promising with regard to throughput-upscale studies to obtain consistent filament quality. Furthermore, these findings are beneficial for formulation development since process optimization and scale-up might be accelerated. Although experiments were performed for only one formulation, the results are likely to be applicable to other formulations.

#### 3.2.5. Impact on Critical Quality Attributes of Filaments and 3D-Printed Dosage Forms

In the last part of the study, the impact of diameter variations on the CQAs printability and uniformity of mass of printed dosage forms was analyzed. Self-produced filaments (T1, T5–T8; screw speeds = 20, 30, 40, 50 and 60 rpm) with different extents of diameter variation were used. These filaments were chosen since it was found that with increasing screw speed and, consequently, lower barrel filling degrees, diameter variations increased (3.2.3). Therefore, a pronounced impact on CQAs of filaments and printed dosage forms was expected. The mechanical resilience of filaments was defined as the main factor for conveyance inside the print head and thus printability [4,5,29]. Therefore, the influence of diameter variations on the YM and distance at break as measures for the mechanical resilience was determined in a tensile test and 3PBT according to Korte and Quodbach [4].

Figure 8 (left) revealed that the YM was hardly influenced. Although there was a significant difference between the YM of filaments produced at 20 rpm and 60 rpm (*p* << 0.01), the effect was considered to be of little practical relevance, as the differences are small (Table 3) and do not influence printability in any way. The impact of screw speed (or SFL) on the mean value of the distance at break was also negligible (Table 3). However, the variation of the distance at break differed distinctly.

Thin sections were less resilient and tended to break faster compared to thicker sections. This was particularly visible when the CoV_distance at break_ was calculated (Figure 8, right). Higher screw speeds increased the CoV_distance break_ from 3% to 20%. This has to be considered, especially when brittle formulations are used. In this case, the reliability of printing is likely to be reduced due to frequently occurring filament breakage inside the print head.

In FDM^TM^, the mean filament diameter is used for the calculation of the feed rate in the G-code. Therefore, it was expected that diameter variations would influence the mass variation of printed dosage forms, as diameter deviations are not considered. The influence of diameter fluctuations due to different screw speeds in extrusion on the mass variation of 20 printed test geometries (3 × 15 × 7.5 mm, Figure 2) was tested.

Additionally, the obtained printed objects were assessed according to Ph.Eur. 2.9.5 “Uniformity of mass of single dose preparations”. In Figure 9 (left), the results of the mean mass of printed geometries from filaments produced at different screw speeds are depicted. The obtained masses of dosage forms printed from the self-extruded formulations were similar, with a slight trend to higher masses from filaments produced at higher screw speeds. Test objects printed from filaments produced with 20 and 30 rpm (T5 and T6) were printed on different days. Due to a necessary reassembly of the print head, a different gap width between the conveyer gears might have been set, leading to different feeding properties and deviating mean masses.

Figure 9 (right) reveals that a higher screw speed causes higher mass variation of printed objects. Screw speeds above 40 rpm led to CoV_mass_ between 6% and 7%, which was not acceptable to ensure content uniformity of printed DDS. According to Ph.Eur. 2.9.5, only one mass may deviate by more than 5% from the mean value. Only dosage forms produced from filaments manufactured at 20 and 30 rpm (T5 and T6; SFL = 0.059 and 0.04) passed the test (Table 3). The other filaments (T1, T7 and T8) did not comply with requirements of the Ph.Eur. For filament formulation T8 (60 rpm), eight dosage forms deviated more than 5%, three of them more than 10%. Filaments T1, T5 and T6 (CoV_diameter_ < 2%) resulted in mass variations of printed objects lower than 4% (20–40 rpm, Table 2).

CoV_mass_ of printed objects from filaments produced with screw speeds of 20 and 30 rpm were similar, even though diameter variations were lower for the filament produced at 20 rpm (T5, CoV_diameter_ = 1.07% vs. 1.76%). An optimization of the print settings might enable an improvement of the quality of printed dosage forms and reveal differences even between low CoV_diameter_ variations [30]. The results of printed objects from commercial PLA filaments revealed that the printer worked at a high precision, enabling the production of even higher quality dosage forms. The study demonstrates that low diameter variations are necessary to obtain drug products with sufficient quality. Compared to objects printed from PLA filaments with a mass variation of approx. 0.5%, the lowest deviation in this study, 3.3% (T5, 20 rpm), is higher. Still, objects printed from T5 were in accordance with the quality requirements of the Ph.Eur.

## 4. Conclusions

A rational approach for HME process analysis and optimization was presented, enabling representative and reproducible quality of the extruded material. A winder was implemented in a TSE manufacturing line as suitable tool for the continuous production of pharma-grade filaments. The results revealed a significant impact of HME parameters on the CQAs of filaments and printed dosage forms. Diameter variations were observed depending on inline-determined material pressure data. In future, monitoring of the material pressure, using a higher sampling rate, correlated with the melt rheology behavior of formulations could be used for inline control of filament diameter.

In the process window, an increase of the PFR from 5 to 10 g/min did not cause high material pressure fluctuations and thus diameter variations. An increase of the screw speed from 20 to 60 rpm led to a distinct increase in diameter inhomogeneity. Above 30 rpm, a high variation of the distance at break of filaments and inacceptable mass variation of the printed dosage forms occurred.

These results highlight the SFL as key parameter for diameter uniformity independently of the set PFR–screw speed combination. A SFL > 0.04 led to filaments with acceptable diameter quality to obtain dosage forms, which fulfilled the requirements of the Ph.Eur. regarding uniformity of mass. A high SFL enabled a homogenous melt transport along the barrel, whereas at lower SFL (<0.03) the material transport was more pulsatile. These findings are especially promising for scale-up of filament production. Optimized process settings for the formulation at an SFL of 0.059 could be identified, which may also applicable to other formulations. Simplified process development for new formulations might be possible based on these results.

Furthermore, the study illustrates the relevance of a suitable diameter control to obtain a full picture of the filament. In the past, diameter measurements were mainly performed offline via a caliper, and undesired filament parts were discarded, which is not suitable for application.

The importance of rational process development and process understanding to obtain high-quality filaments to fulfill required product CQAs was shown. The findings are an improvement in the direction of continuous processing and quality control of filaments, as well as printed dosage forms.

## Figures and Tables

**Figure 1 pharmaceutics-12-00511-f001:**
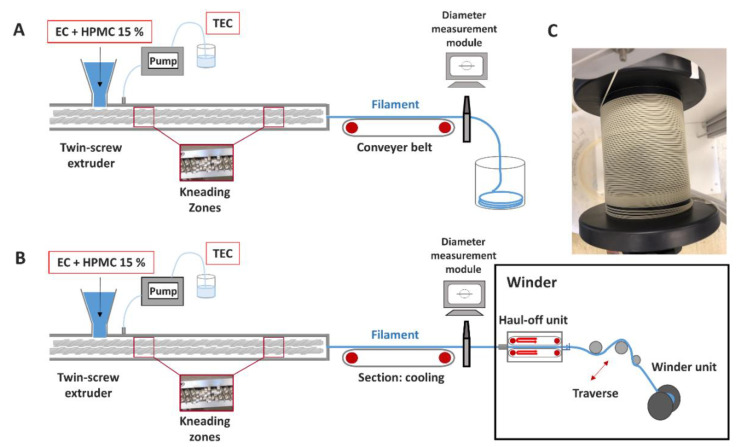
HME setup for filament production before (**A**) and after (**B**) optimization by implementing a winder. Subfigure (**C**) displays an exemplary winding process in the winder unit.

**Figure 2 pharmaceutics-12-00511-f002:**
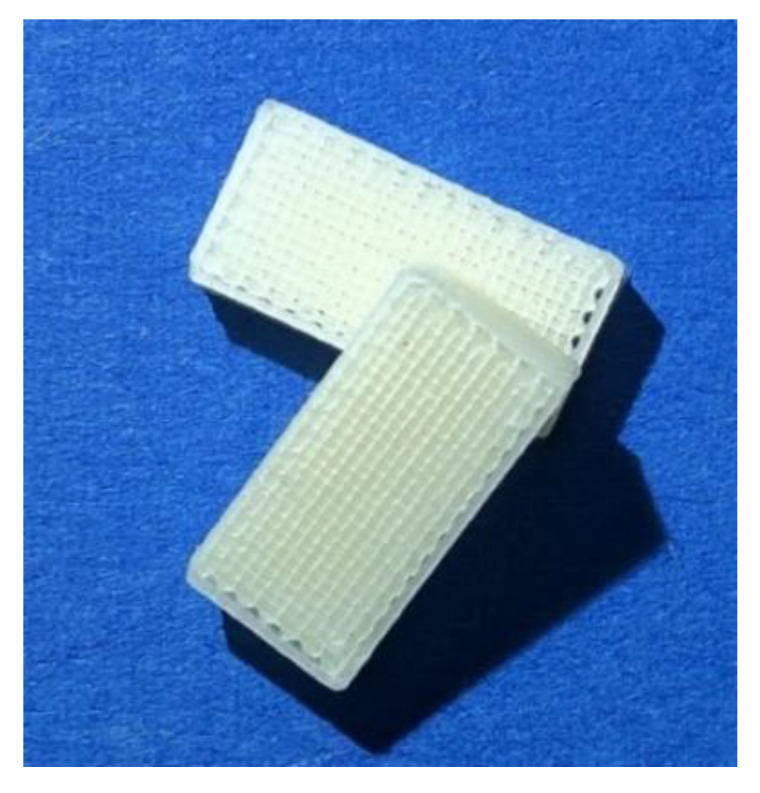
Picture of printed objects (3 × 15 × 7.5 mm) using in-house-produced filaments based on EC and HPMC for the evaluation of uniformity of mass according to Ph.Eur. 2.9.5.

**Figure 3 pharmaceutics-12-00511-f003:**
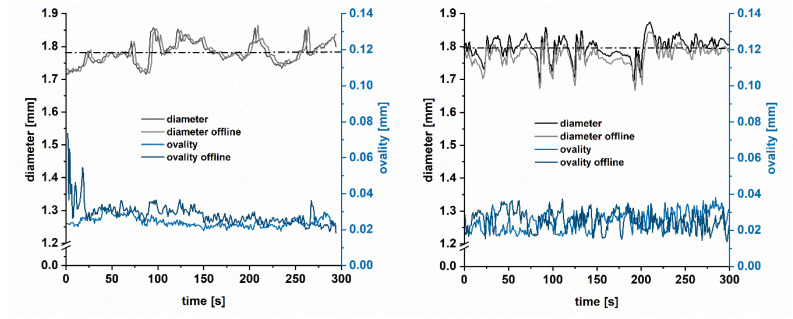
Inline- and offline-determined diameter and ovality (∆Ø_max/min_) as a function of time (1 Hz) for each applied PFR = 5 g/min (**left**); 10 g/min (**right**). The dashed line marks the mean value of the inline-determined diameter.

**Figure 4 pharmaceutics-12-00511-f004:**
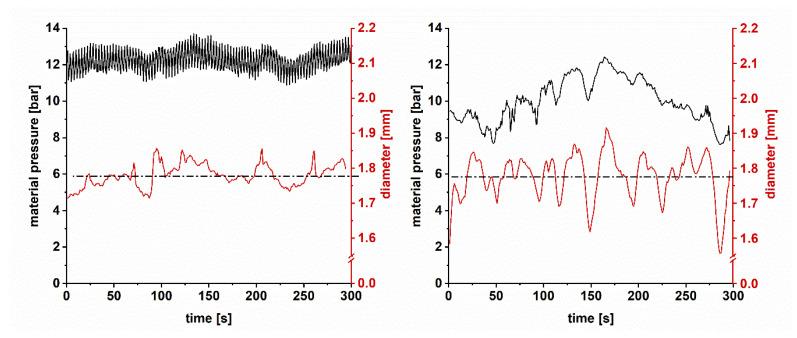
Inline determined material pressure and diameter as a function of time, where a PFR of 5 g/min and a screw speed of 40 rpm (**left**) or 60 rpm (**right**) were applied. The dashed lines mark the mean diameter.

**Figure 5 pharmaceutics-12-00511-f005:**
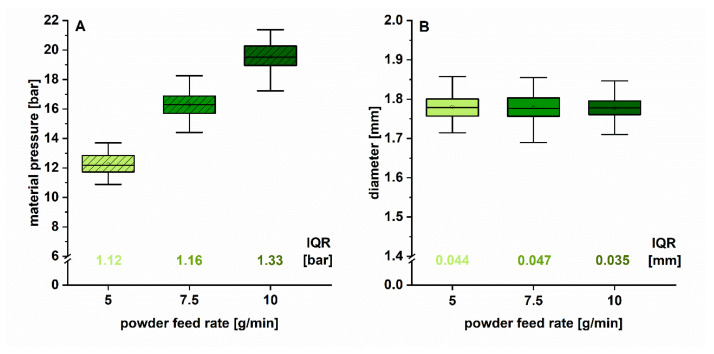
Boxplot of inline measured material pressure at the die (**A**) and filament diameter (**B**) (1 Hz, *n* > 294) for different applied PFR. The respective IQRs (box width) are indicated.

**Figure 6 pharmaceutics-12-00511-f006:**
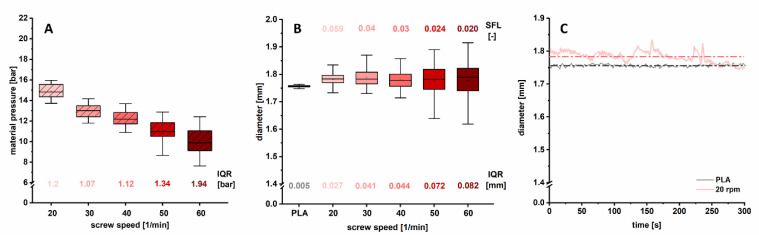
Boxplot of material pressure (**A**) and filament diameter (**B**) for different applied screw speeds during extrusion (1 Hz; *n* > 290). Resulting SFL and IQRs (box width) are indicated. Inline determined diameter as function of time for commercial PLA filament compared to produced filament with 20 rpm (**C**); dashed lines mark the mean value.

**Figure 7 pharmaceutics-12-00511-f007:**
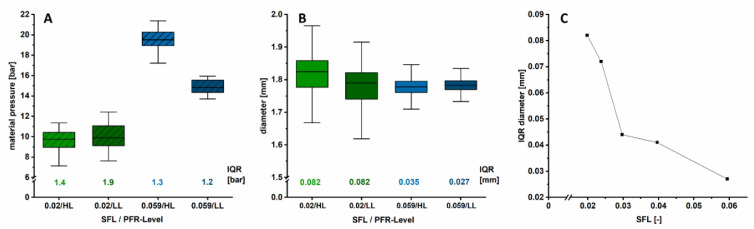
Boxplots of inline determined material pressure (**A**) and diameter (**B**) during HME for two different applied SFL on a high (10 g/min, HL) and low (5 g/min, LL) PFR-level, respectively (1 Hz, *n* > 290). IQRs of pressure and diameter are indicated. (**C**) displays diameter IQR as function of different applied screw speeds (*n* > 290).

**Figure 8 pharmaceutics-12-00511-f008:**
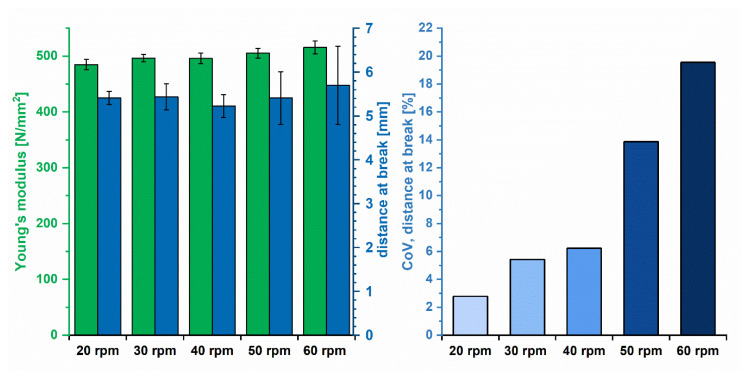
Determined YM and distance at break of filaments produced at different screw speeds (mean ± CI, *n* = 6 (α = 0.05), **left**) and the CoV_distance at break_ related to the applied screw speed (**right**).

**Figure 9 pharmaceutics-12-00511-f009:**
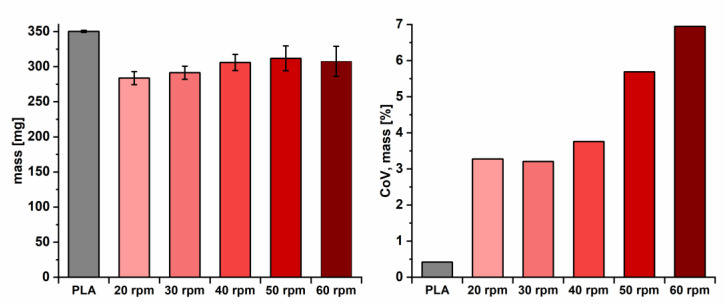
Mass of printed objects (*n* = 20, mean ± s) of filaments produced with different screw speeds compared to commercial PLA filaments (**left**) and CoV_mass_ (**right**).

**Table 1 pharmaceutics-12-00511-t001:** Overview of experimental plan (SFL = specific feed load).

Trial	Batch Name	PFR [g/min]	LFR [g/min]	Screw Speed [1/min]	SFL [–]	Haul-off Speed [m/min]
**Winder implementation**	T1	5	0.56	40	0.03	2.0
	T2	10	1.11	40	0.059	4.1
**Influence of PFR**	T1	5	0.56	40	0.03	2.0
	T3	7.5	0.83	40	0.045	3.0
	T4	10	1.11	40	0.059	4.0
**Influence of screw speed**	T5	5	0.56	20	0.059	2.0
	T6	5	0.56	30	0.04	2.0
	T1	5	0.56	40	0.03	2.0
	T7	5	0.56	50	0.024	2.0
	T8	5	0.56	60	0.02	2.0
**Influence of SFL**	T5	5	0.56	20	0.059	2.0
	T8	5	0.56	60	0.02	2.0
	T4	10	1.11	40	0.059	4.0
	T9	10	1.11	120	0.02	4.0

**Table 2 pharmaceutics-12-00511-t002:** Inline determined diameter data of produced filaments and commercial PLA (mean, standard deviation (s) and coefficient of variation (CoV), *n* = ~300).

Diameter	T1	T2	T3	T4	T5	T6	T7	T8	T9	PLA
**Mean [mm]**	1.781	1.798	1.779	1.778	1.782	1.789	1.776	1.777	1.821	1.756
**s [mm]**	0.032	0.031	0.035	0.033	0.019	0.032	0.054	0.064	0.062	0.004
**CoV [%]**	1.80	1.72	1.98	1.83	1.07	1.76	3.06	3.63	3.38	0.21

**Table 3 pharmaceutics-12-00511-t003:** Overview of the impact of diameter variations (related to applied screw speeds) on mechanical resilience of filaments and mass uniformity of printed test geometries, as well as results according to Ph.Eur. 2.9.5 (green numbers = corresponded; red numbers = not corresponded; ND = not determined).

Sample	Mass of Printed Dosage Form [mg] (Mean ± s, *n* = 20),	Ph.Eur. 2.9.5		Mechanical Properties	
		*n* > 5%	*n* > 10%	YM [N/mm^2^] Mean ± CI (α = 0.05), *n* = 6	Distance at Break [mm], Mean ± CI, (α = 0.05), *n* = 6
**PLA**	350 ± 1.5	-	-	ND	ND
**T5, 20 rpm**	283.8 ± 9.3	1	-	484.7 ± 9.4	5.408 ± 0.151
**T6, 30 rpm**	291.4 ± 9.2	1	-	496.5 ± 6.5	5.433 ± 0.295
**T1, 40 rpm**	306.1 ± 11.5	3	1	496.0 ± 9.7	5.225 ± 0.261
**T7, 50 rpm**	311.9 ± 17.7	7	1	505.5 ± 8.7	5.408 ± 0.600
**T8, 60 rpm**	307.5 ± 21.4	8	3	515.5 ± 11.4	5.700 ± 0.892

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
