# Peer review of "Hot-Melt Extrusion Process Fluctuations and Their Impact on Critical Quality Attributes of Filaments and 3D-Printed Dosage Forms"

_pharmaceutics, 2020, doi:10.3390/pharmaceutics12060511_

Round 1

Reviewer 1 Report

The present paper reports on the evaluation of two process parameters, e.g. powder feed rate and screw speed, to prepare ethylcellulose/hydroxypropylcellulose filaments by hot melt extrusion. These filaments were intended to prepare solid dosage forms. The evaluated filaments attributes were shape (diameter, ovality) and mechanical properties (Young’s modulus and distance at break); then, the printed pararallelepipeds were evaluated for mass. The work is interesting and almost soundly performed. The experimental were designed by one factor at time (OFAT) methodology; probably, the suggested tools by the Quality by Design paradigm should evaluate in depth the critical process parameters for the next steps of the research work. Therefore, in my opinion, it should be accepted according to some minor revisions:

- Abstract: the Authors stated that ‘… the impact of diameter homogeneity on … dose accuracy of printed dosage forms were investigated’. Probably, it should be better to write mass uniformity of printed objects or of placebo dosage forms’

- Introduction: the authors stated that ‘The diameter consistency influences the critical quality attributes (CQAs) of filaments …’. Is the diameter consistency a CQA?

- The authors should check the number of figures in Tables 1 and 3.

- Table 1: the authors should check the values of the Haul-off speed in the Trials 4 and 9 (T4 and T9).

- In the Chapter 2.2.6. a typo occurred: United States require initials capital letters

- Chapter 3.1: the Authors should state if it is significant the difference between the ovality after the winding process and the inline determined ovality

- In the Chapter 3.2.1. title a typo occurred: ‘und’ was used instead of ‘and’

- In the Chapter 3.2.4. the authors wrote ‘Bothe …’, is it a typo?

- Chapter 3.2.4.: the Authors stated that ‘a faulty calibration of the pressure probe at different temperatures’ occurred. Is it possible to fix this problem?

- Chapter 3.2.5.: the Authors should justify the choice of T1, T5-T8 filaments to prepare dosage forms.

- Chapter 3.2.5.: the Authors stated that ‘Figure 8 (left) revealed that YM was hardly influenced.’ Should the authors explain better this sentence? Are the differences among YM considerable and/or significant?

- Chapter 3.2.5.: should the authors explain the mass values trend when screen speed increases?

- Table 3: are the names of the Sample correct?

Author Response

Abstract: the Authors stated that ‘… the impact of diameter homogeneity on … dose accuracy of printed dosage forms were investigated’. Probably, it should be better to write mass uniformity of printed objects or of placebo dosage forms’

We thank you very much for this comment. The sentence was adjusted to reflect your comment: “In addition, the impact of diameter variation on the critical quality attributes of filaments (mechanical properties) and uniformity of mass of printed drug-free dosage forms was investigated.”  (l 18)

Introduction: the authors stated that ‘The diameter consistency influences the critical quality attributes (CQAs) of filaments …’. Is the diameter consistency a CQA?

You are right with your comment. The diameter consistency is one of the most important CQA to consider, in order to guarantee printability of filaments, uniformity of mass as well as resolution of printed objects.

The sentence was changed to clarify the stated comment: The filament diameter is a critical quality attribute (CQA) to consider, which is strongly related to the process parameters during HME and filament collection process [4]. Not only the diameter itself but also the diameter consistency is an important CQA, influencing the quality of filaments and printed dosage forms [10]. Particularly noteworthy factors that are influenced are the printability of filaments, uniformity of mass, content uniformity as well as resolution of printed dosage forms.” (l 37)

The authors should check the number of figures in Tables 1 and 3.

Thank you for the advice, this was an error and corrected.

Table 1: the authors should check the values of the Haul-off speed in the Trials 4 and 9 (T4 and T9).

Thank you for pointing out the mistake. The correct values have been implemented. At the section “Influence of SFL”: the haul-off speed values of T4 and T9 was changed from 2.0 to 4.0 m/min. (l 119)

In the Chapter 2.2.6. a typo occurred: United States require initials capital letters

The typo was corrected.

Chapter 3.1: the Authors should state if it is significant the difference between the ovality after the winding process and the inline determined ovality

The p-values were added. Although a significant difference between the ovalities (inline/offline) was found (5g/min), it was considered not relevant as it was almost certainly caused by the movements due to manual feeding (less tension) during offline measurement. We added this also to the manuscript.

“The ovality after the winding process (0.028 ± 0.005 mm) is significantly higher compared to the inline determined ovality (0.024 ± 0.003 mm; p << 0.01), which is very likely a measurement artifact caused by a lopsided filament position inside the laser unit, as it was not under full tension during offline feeding.“ (l 199)

In the Chapter 3.2.1. title a typo occurred: ‘und’ was used instead of ‘and’

Thank you. This was corrected.

In the Chapter 3.2.4. the authors wrote ‘Bothe …’, is it a typo?

Also corrected.

Chapter 3.2.4.: the Authors stated that ‘a faulty calibration of the pressure probe at different temperatures’ occurred. Is it possible to fix this problem?

Yes, it can be avoided. Normally, the pressure probe calibration is performed at process temperature. Since both processes were conducted on different days and the values were evaluated afterwards, we can only assume that in one of the pressure calibrations was accidently performed at room temperature, causing the stated differences.

Chapter 3.2.5.: the Authors should justify the choice of T1, T5-T8 filaments to prepare dosage forms.

Thank you for the useful advice. The justification of filament selection for the investigation of impact on CQAs was done more precisely. The increase of screw speeds and consequently lower barrel filling degree led to higher diameter variations.  Thus, filaments were selected as an influence on mechanical properties and mass uniformity of printed dosage forms was expected.

“Self-produced filaments (T1, T5 - T8; screw speeds = 20, 30, 40, 50 and 60 rpm) with different extents of diameter variation were used. These filaments were chosen, since it was found that with increasing screw speed and consequently lower barrel filling degrees, diameter variations increased (3.2.3). Therefore, a pronounced impact on CQAs of filaments and printed dosage forms was expected.” (l 369)

Chapter 3.2.5.: the Authors stated that ‘Figure 8 (left) revealed that YM was hardly influenced.’ Should the authors explain better this sentence? Are the differences among YM considerable and/or significant?

Thank you, for the advice. The YM-changes dependent on applied screw speed were explained in more detail and a statistical test was performed (p- value between to extremes was calculated and added).

Although there was a significant difference between the YM of filaments produced at 20 rpm and 60 rpm (p << 0.01), the effect was considered to be of little practical relevance, as the differences are small (Table 3) and do not influence printability in any way. The impact of screw speed (or SFL) on the mean value of the distance at break was also negligible (Table 3).” (l 383)

Chapter 3.2.5.: should the authors explain the mass values trend when screen speed increases?

We added a brief explanation about this issue.

“The obtained masses of dosage forms printed from the self-extruded formulations were similar, with a slight trend to higher masses from filaments produced at higher screw speeds. Test objects printed from filaments produced with 20 and 30 rpm (T5 and T6) were printed on different days. Due to a necessary reassembly of the print head, a different gap width between the conveyer gears might have been set, leading to different feeding properties and deviating mean masses.” (l 400)

Table 3: are the names of the Sample correct?

As stated above, this was an error and corrected.

Reviewer 2 Report

In the current article the authors have employed HME processing for the manufacturing of filaments and investigated the effect of process fluctuations on the CQAs of 3D printed dosage forms.  This is a very interesting well organised study that provides new insights to those who work on FDM technology. 

I ma very impressed with the attention to detail and the plethora of characterisation techniques that the authors have use. The discussion is in agreement with the experimental findings.  I believe the article will attract the interest of the journal's readership and I recommend it for publication. 

Author Response

In the current article the authors have employed HME processing for the manufacturing of filaments and investigated the effect of process fluctuations on the CQAs of 3D printed dosage forms.  This is a very interesting well organised study that provides new insights to those who work on FDM technology. 

I ma very impressed with the attention to detail and the plethora of characterisation techniques that the authors have use. The discussion is in agreement with the experimental findings.  I believe the article will attract the interest of the journal's readership and I recommend it for publication. 

We would like to extend our thanks to the reviewer for his kind words about our manuscript.